# Comparative Analysis of Microbial Communities in Diseased and Healthy Sweet Cherry Trees (*Prunus avium* L.)

**DOI:** 10.3390/microorganisms12091837

**Published:** 2024-09-05

**Authors:** Tong Zhou, Xiaojuan Huang, Danyang Zhu, Yan Tang, Hongli Xu, Fanrong Ran, Hasin Ullah, Jiangli Tan

**Affiliations:** Shaanxi Key Laboratory for Animal Conservation in Western China, College of Life Sciences, Northwest University, 229 North Taibai Road, Xi’an 710069, China; zhoutong@stumail.nwu.edu.cn (T.Z.); huangxiaojuan@stumail.nwu.edu.cn (X.H.); zhudanyang@stumail.nwu.edu.cn (D.Z.); tangyan1@stumail.nwu.edu.cn (Y.T.); 202233084@stumail.nwu.edu.cn (H.X.); 202133373@stumail.nwu.edu.cn (F.R.); hasenullah888@yahoo.com (H.U.)

**Keywords:** Illumina sequencing, microbial community, *Prunus avium* (L.), tumor gummosis

## Abstract

The European sweet cherry *Prunus avium* (L.), a member of the Rosaceae family, is one of the most popular and economically valuable fruits. However, the rapid spread of gummosis and poor management practices have become the major obstacles to their production. To identify pathogenic microorganisms responsible for gummosis disease, we conducted observations comparing the garden of Bailuyuan, which heavily suffered from gummosis disease and horn beetle damage, with the orchard of Mayuhe, which only suffered from gummosis disease, both from Xi’an, Shaanxi, China. Samples were obtained from the healthy tissues and gummosis disease tissues that used the Illumina sequence of 16S rRNA and the internal transcribed spacer region (ITS) to identify bacterial and fungal communities in these samples. An alpha diversity analysis revealed a significantly higher fungal diversity of disease than in healthy tissue in the gummosis period. The results suggested that an imbalance in the fungal genera may be associated with gummosis disease. Species relative analyses showed some bacterial genera (*Pelagibacterium*, *Halomonas*, *Azospirillum*, *Aquabacterium* and *Alistipes*) and fungal genera (*Penicillium*, *Alternaria* and *Rhodotorula*) in the diseased tissues of gummosis. Among these, the increased relative abundance of the bacteria genes *Halomonas*, *Pelagibacterium*, *Chelativorans*, *Pantoea*, *Aquabacterium*, *Alternaria* and fungi genes *Penicillium*, *Cystobasidium*, *Rhodotorula* may be associated with gummosis of *P. avium*. The bacterial genera *Methylobacterium*, *Psychroglaciecola*, *Aeromonas*, *Conexibacter* and fungal genera *Didymella*, *Aureobasidium*, *Mycosphaerella*, *Meyerozyma* are probably antagonists of the pathogen of gummosis. These findings are an initial step in the identification of potential candidates for the biological control of the disease.

## 1. Introduction

The European sweet cherry *Prunus avium* (L.), a member of the Rosaceae family, as one of the most popular fruits, originating from West Asia and Southeast Europe, was introduced in 1871 [1] and is cultivated on more than 233,300 hectares, yielding approximately 1.2 million tons per year recently in China with high economic value [2]. However, as the single planting area of sweet cherry trees increases, the occurrence of various diseases and pests is also becoming more and more serious [3]. Among them, cherry gummosis syndrome with symptoms of exuding large amounts of gum on the trunk and branches, accompanied by sunken lesions on the bark, has become a major obstacle to production [4,5]. The necrotizing area extends to the xylem to form black to brown staining of the tissue [4]. Zhang et al. [6,7] reported that disease rate over 70% of sweet cherry trees have been infected in Shandong and Sichuan. The rapidly spread disease could damage the cherry orchards seriously [7,8], and Zhang et al. [9] showed that sweet cherry gummosis was found in more than a 10% incidence of branches in an experimental orchard in Shanxi Province. The situation became even worse in Bailuyuan, a modern agricultural demonstration area of Xi’an Shaanxi in recent years. Thousands of sweet cherry trees suffered from the serious problem with symptoms of gummosis, bark burst and severe tumors on the trunk and branches named by the local people as “black knot disease”, and this often led to the death of trees [10]. Current treatments, such as frequent scraping of the lesions and painting of the exposed tissue with chemicals to inhibit the pathogens in the trees are often ineffective [11,12]. It is important to determine the cause of the disease and develop efficient strategies for synthetic control.

The syndrome is frequently observed in various deciduous fruit tree species such as plums [13,14], peaches [15], *Prunus armeniaca* [16], nectarines [17] and cherries [9]. The causal agent is unknown in most cases, but the symptoms are very similar [4]. The fungal gummosis syndrome of deciduous fruit trees was described in the United States and Japan in the 1960s as caused by *Botryosphaeria dothidea* [18]. Al-Sadi reported that *Lasiodiplodia hormozganensis* and *Neoscytalidium dimidiatum* were the causal agents of the gummosis syndrome on sweet limes in Oman [19]. *Botryosphaeria dothidea* has been reported as the causal agent of gummosis syndrome in Japanese apricot in China [20]. However, more and more investigation results support that the factors leading to gummosis are complex. Gummosis is classified into physiological gummosis and invasive gummosis, respectively, which are caused by insect pests, wounds, osmotic stress and pathogenic bacteria [21,22,23]. For instance, Zhang et al. [9] offered the first report of the fungus *Botryosphaeria dothidea* causing gummosis of *Prunus avium* L. (sweet cherry). Li, Akbaba and Marronin suggested that the pathogen of cherry gummosis is the bacteria *Pseudomonas syringae* [24,25,26]. Cherry trees growing in exposed or poorly drained sites, or shallow soil, may exhibit gummosis that is caused by the bacteria *Pseudomonas* and the fungus *Schizophyllum* [27]. Gummosis in shoots of *Prunus yedoensis* was studied in relation to hormonal status, leaf abscission and chemical composition of gums [28]. Li et al. [10] comparatively analyzed microbial communities in the rhizosphere and tissues of diseased and healthy cherry trees from Bailuyuan using Illumina sequencing of 16S rRNA gene and the internal transcribed spacer (ITS) amplicon. The results showed that fungal diversity in diseased tissue was significantly higher than in healthy tissue, an imbalance in fungal flora may be related to black spot disease. Although gummosis was discovered [9], the mechanism of this disease of the sweet cherry is still not well understood.

In recent years, our research group found that the sweet cherry in Bailuyuan (loess plateau) affected by “black knot disease” were also severely damaged by the stem borer, the red-necked longhorn beetle *Aromia bungii* [29] (Coleoptera: Cerambycidae). Interestingly, only 57.5 km away from Bailuyuan, in the orchard of Mayuhe (sandy soil, near the bank of Ganyu river) which also at the northern piedmont of the Qin Mountains, the sweet cherry tree suffered with gummosis syndrome, but without the longhorn beetle *Aromia bungii* damage, and it did not have tumors. Obviously, by comparing the microbial communities of the diseased tissues from these two areas, it is possible to make the main cause clear. The objective of this study used Illumina sequencing technology to investigate the differences between bacterial and fungal communities in healthy and diseased tissues of the sweet cherry.

## 2. Materials and Methods

### 2.1. Experimental Material

Samples were collected from cherry plantations located in Bailuyuan, Baqiao District, Xi’an, Shaanxi Province, China (E 109.12, N 34.19, alt. 706.34 m, loess plateau) and Mayuhe, Jiang Village, Huyi District, Xi’an, Shaanxi Province, China (E 108.51, N 34.05, alt. 507.75 m, sandy soil), following the method described by Ding et al. [10]. Sterile equipment was used to observe and remove tissues from trees, which were then placed in polyethylene bags. A five-point sampling method was employed to collect inner bark tissues with tumor gummosis and healthy tissues. Five samples were collected and combined into one sample for each condition, which were then ground and passed through a 2 mm sieve. The samples were placed in sterile containers, stored at 4 °C and DNA extraction was performed within 48 h. Sequencing was completed by Personalbio, Shanghai, and microbiome statistics for gummosis and healthy tissues from the two sampling locations were obtained from these data (Figure 1).

### 2.2. DNA Extraction, PCR Analysis and MiSeq Illumina Sequencing

Total microbial genomic DNA was extracted from 200 mg of the healthy tissues and each gummosis tissue within the sweet cherry tree. Total genomic DNA samples were extracted using the OMEGA Soil DNA Kit (D5625-01) (Omega Bio-Tek, Norcross, GA, USA), following the manufacturer’s instructions. The concentration and purity of DNA were determined using a NanoDrop Spectrophotometer (Thermo Fisher Scientific, Waltham, MA, USA), and DNA samples were stored at −80 °C until further processing.

Total bacterial 16S rDNA was quantified by PCR. Primer pairs for the bacteria were the forward primer 338F (5′-ACTCCTACGGGAGGCAGCA-3′) and the reverse primer 806R (5′-GGACTACHVGGGTWTCTAAT-3′), targeting the 16S V3V4 region. The extracted DNA was amplified with two-step PCR, with sample-specific 16 bp bar codes incorporated into the forward and reverse primers for multiplex sequencing. Amplification was carried out using the following program: 95 °C for 2 min for the initial denaturation and then 25/10 cycles (for the first and second amplification step, respectively) of 98 °C for 30 s followed by annealing at 55 °C for 30 s and extension at 72 °C for 90 s, with a final extension of 5 min at 72 °C. Amplicon libraries for the fungi were prepared using an identical approach, using primers ITS5 and ITS2 [30], which amplify the internal transcribed spacer (ITS) region. The samples were sequenced (PE250) on an Illumina MiSeq platform, provided by Shanghai Personal Biotechnology Co, Ltd. (Shanghai, China), according to the manufacturer’s instructions. Overall, PCR amplicons were purified with Agencourt AMPure Beads (Beckman Coulter, Indianapolis, IN, USA) and quantified using the PicoGreen dsDNA Assay Kit (Invitrogen, Carlsbad, CA, USA).

### 2.3. Sequencing Data Processing

Raw Illumina FASTQ files were quality filtered, denoised, merged and chimera removed using the Quantitative Insights into Microbial Ecology and DADA2 plugin (QIIME2 v 2019.4, https://docs.qiime2.org/2019.4/tutorials, accessed on 29 January 2021) [31,32]. All libraries had been denoised, the ASVs feature sequences and ASV tables were merged and singleton ASVs were removed. The RDP FrameBot (version 1.2, Michigan State University, East Lansing, MI, USA) [33] that was based on the seed protein sequences of the corresponding functional genes (https://github.com/rdpstaff/Framebot, accessed on 30 March 2021) was used to correct insertion and deletion errors. The supplemented reference sequences were protein sequences that passed the checksum, met specific requirements and used de novo mode, an amino acid length filtering threshold of 50 and all other parameters of default values. Non-singleton complication sequence variants (ASVs) were aligned with MAFFT (version 7, Kyoto University, Kyoto, Japan) [34] and used to construct a phylogeny with FastTree2 (version 2.1.10, University of California, Berkeley, CA, USA) [35]. The feature sequences of each ASV were compared using the Silva database [36] for bacterial 16S rRNA genes and the UNITE database [37] for fungal ITS sequences. Species annotation was performed in QIIME2 (version 2019.4, Arizona State University, Tempe, AZ, USA) [38] using a per-trained Naive Bayes classifier with default parameters.

### 2.4. Functional Gene Prediction

Phylogenetic investigation of communities by using reconstruction of unobserved states (PICRUSt, version 1.1.3, University of British Columbia, Vancouver, BC, Canada) is a bioinformatics tool that uses 16S ribosomal DNA sequences to predict the functional gene content of microorganisms [39]. In the present study, we used PICRUSt to obtain an overview of the genomic and metabolic features represented by the adherent bacterial communities in our samples. We associated ASVs with known bacterial genomes precalculated in PICRUSt by first picking closed ASVs against the Greengenes 16S rRNA gene database using QIIME (v1.7.0, University of Colorado Boulder, Boulder, CO, USA) [31]. The resulting ASVs table was then normalized using the script normalize_by_copy_number.py and used for metagenome inference of Kyoto Encyclopedia of Genes and Genomes (KEGG) Orthologs using PICRUSt.

### 2.5. Statistical Analysis

The analyses of the sequence data were mainly performed with QIIME2 (version 2019.4, Arizona State University, Tempe, AZ, USA) and R packages (v3.2.0, R Core Team, Vienna, Austria). ASV-level alpha diversity indices, such as Chao1 richness estimator, observed species, Shannon diversity index, Simpson index, Faith’s PD, Pielou’s evenness and Good’s coverage were calculated using the ASV table in QIIME2 and visualized as box plots. ASV-ranked abundance curves were generated to compare the richness and evenness of ASVs among samples. The significance of the differentiation of microbiota structure among groups was assessed by PERMANOVA (Permutational multivariate analysis of variance) [40], and ANOSIM (Analysis of similarities) [41,42]. Permdisp [43] used QIIME2. The taxonomy compositions and abundances were visualized using MEGAN (version 6.21.1, Institute of Computer Science, Humboldt-University of Berlin, Berlin, Germany), [44] and GraPhlAn (version 0.99.7, Department of Bioinformatics, University of Rome “La Sapienza”, Rome, Italy), [45]. A Venn diagram was generated to visualize the shared and unique ASV among samples or groups using the R package “VennDiagram” (version 1.6.20, RStudio, Boston, MA, USA), based on the occurrence of ASVs across samples/groups regardless of their relative abundance [46]. Microbial functions were predicted by using PICRUSt2 (Phylogenetic investigation of communities by reconstruction of unobserved states) upon MetaCyc (http://metacyc.org, accessed on 21 April 2021) and the KEGG (https://www.kegg.jp, accessed on 21 April 2021) database.

## 3. Results

### 3.1. Amplicon Sequencing and Community Diversity Overview

Based on the number of sequences sampled compared to the ASVs, we were able to generate dilution curves showing that the results of bacteria sequenced with 16S rRNA and fungi sequenced with ITS steadily increased and eventually reached a plateau. This indicates that the amount of sequencing data obtained is sufficient to cover the most tested samples of bacterial and fungal taxa. Therefore, examining the amount of the study of sequencing can reflect the composition and types of bacterial and fungal communities examined (Figure 2).

A total of 664,789 valid bacterial reads and 398,925 valid fungal reads were obtained from all the samples. The mean lengths of the bacterial and fungal sequences were about 307.0 bp and 265.9 bp, respectively. The minimum read lengths of bacterial and fungal sequences were 129 bp and 154 bp, while the maximum read lengths were 432 bp and 362 bp, respectively. The total number of ASVs (groups of organisms that share at least 100% rRNA gene sequence identity) of bacteria and fungi was high in this study. The number of different bacterial ASVs in all samples ranged from 502 to 2240, and the number of the fungal ASVs ranged from 116 to 124 (Appendix A). The results of alpha diversity analysis showed that there were no significant differences in the abundance and diversity of bacteria and fungi in samples J1, J2 and B1, B2 (J3, J4 and B3, B4). Simpson, Shannon, and Pielou_e index values of bacteria in Z1 were approximately higher than B3, and fungi in B2 were slightly higher than B4. Chao1, Good coverage, and Observed species of the bacteria in B3 were higher than in B1, and the fungi in B4 were higher than in B2 (Table 1).

### 3.2. Changes in Bacterial Community Structure at Phylum and Genus Level

A Venn diagram (Figure 3) was generated to evaluate the distribution of ASVs among the different samples. This analysis revealed that 2742 ASVs (91.95%) were shared between J1, J3 and B1, B3, and 240 ASVs (8.05%) were shared between J2, J4 and B2, B4. The distribution of ASVs demonstrates that the bacterial communities of B1/B3 and B2/B4 are substantially different. It is noteworthy that 138 ASVs of bacterial communities were shared between B1 and B3, and 161 and 1016 ASVs of bacteria were found exclusively in B1 and B3 in Figure 3a. Meanwhile, it is noteworthy that of the fungal communities, 59 ASVs were shared between B2 and B4 (Figure 3b).

### 3.3. Changes in Bacterial Community Structure at the Phylum and Genus Level

During the tumor gummosis phase, a total of 183 bacterial species were identified, belonging to 14 phyla and 173 genera (Appendix A). The dominant phyla were Proteobacteria (42.1%), Firmicutes (37.0%), Bacteroidetes (11.8%), Actinobacteria (5.8%), Fusobacteria (2.1%), Cyanobacteria (0.4%), Chloroflexi (0.4%), Acidobacteria (0.1%), Tenericutes (0.1%) and Epsilonbacteraeota (0.1%) based on average relative abundance. In addition, the relative abundance of Proteobacteria in J1 was 51.5% but only 32.8% in B1. Similarly, the relative abundance of Actinobacteria J1 was 8.3%, much higher than that in B1 (3.2%), and the relative abundance of Fusobacteria, Cyanobacteria and Chloroflexi J1 was 4.1%, 0.6% and 0.7%, much higher than that in B1 (0%, 0.3% and 0.1%). However, Firmicutes and Bacteroidetes were significantly more abundant in B1 (46.8% and 16.0%, respectively) than in J1 (27.2% and 7.5%, respectively), and Acidobacteria Tenericutes and Epsilonbacteraeota relative abundance (0.3%, 0.2% and 0.1%) was detected only in B1 (Figure 4a). At the genus level, the relative abundances of *Azospirillum*, *Aquabacterium*, *Alistipes*, *Ruminococcaceae_UCG-014*, *Clostridiales_vadinBB60_group*, *Bacteroides* and *Christensenellaceae_R-7_group* were higher in B1 (8.3%, 6.0%, 5.9%, 5.7%, 4.1%, 4.2%, 4.2% and 4.0%, respectively) than in those in J1 (3.3%, 3.7%, 0.7%, 0.7%,2.2%, 1.9% and 1.8%, respectively). *Methylobacterium*, *Inhella* and *Aeromonas* were less abundant in B1 than in J1 (Figure 5a).

During the gummosis phase, 33 bacterial phylum and 547 genera were identified (Appendix A). The dominant phyla were Proteobacteria (74.5%) and Actinobacteria (10.2%), based on average relative abundance (Figure 4a). In addition, the relative abundance of Proteobacteria in B3 was 77.7% but only 71.4% in J3. Similarly, the relative abundance of Bacteroidetes, Gemmatimonadetes and Verrucomicrobia in B3 was 5.5%, 0.9% and 0.5%, substantially higher than that in J3 (0.8%, 0.7% and 0.3%). However, Actinobacteria, Firmicutes, Acidobacteria, Chloroflexi, Planctomycetes, and Deinococcus-Thermus are significantly more abundant in J3 than in B3 (Figure 4a). At the genus level, the relative abundances of *Halomonas*, *Pelagibacterium*, *Chelativorans* and *Pantoea* were higher in B3 than in those in J3 (Figure 5a).

At the phylum level, the Planctomycetes, Gemmatimonadetes, Deinococcus-Thermus and Verrucomicrobia are only present in B3 compared to the identification of tumor gummosis (B1). In addition, the relative abundance of Proteobacteria was 77.7% in B3, but it was only 32.8% in B1. Similarly, the relative abundance of Chloroflexi and Actinobacteria was much higher in B3 (1.6% and 6.2%) than that in B1 (0.1% and 0.3%) (Figure 4a and Appendix A). With the genus level, the relative abundance of *Methylobacterium* in B3 was 3.8%, but it was only 1.4% in B1 (Figure 5a and Appendix A).

### 3.4. Changes in Fungal Community Structure at the Phylum and Genus Level

A total of 240 different fungal ASVs were identified in this study. A total of 124 ASVs (51.67% of all the ASVs) were shared between B2 and J2, and 116 ASVs (48.33% of all the ASVs) were shared between B4 and J4. Overall, 59 fungal ASVs were shared between B2 and B4, and 72 and 92 ASVs were found in B2 and B4, respectively (Appendix A). The dominant fungal phyla in all the fungal samples (B2 and B4) were Ascomycota and Basidiomycota (Figure 4b and Appendix A).

During the tumor gummosis phase, six phylum and 48 genera of fungi were identified (Appendix A). The relative abundance of Ascomycota and Basidiomycota in J2 was 18.3% and 0.2%, but it was only 0.2% and 0.1% in B2. Among genera, the abundance in J2 of *Aureobasidium*, *Alternaria*, *Mycosphaerella*, *Didymella*, *Penicillium*, *Paricillium*, *Paraconiothyrium*, *Cystobasidium*, *Rhodotorula*, *Gibberella* and *Meyerozyma* was higher in B2 (Figure 5b).

In the gummosis phase, 59 fungal genera and four phyla were identified (Appendix A), with Ascomycota and Basidiomycota being highly abundant at the phylum level (Figure 4b). *Penicillium* was the most abundant genus, along with *Aureobasidium*, *Alternaria*, *Didymella*, *Paraconiothyrium*, *Cystobasidium*, *Rhodotorula*, *Gibberella*, *Malassezia*, and *Golubevia* in B4 (Figure 5b).

At the phylum level, compared with B2, B4 had a significantly higher percentage of Ascomycota (80.4%) and Basidiomycota (6.3%). Mortierellomycota and Mucoromycota were not detected in B4. At the genus level, *Penicillium* was significantly more abundant in B4 than B2 with an average relative abundance of 50.5%. *Aureobasidium*, *Alternaria*, *Malassezia*, *Didymella*, *Paraconiothyrium*, *Cystobasidium* and *Rhodotorula* were more abundant in B4 (0.5%, 0.5%, 0.1%, 0.4%, 0.8%, 1.1%, 1.5% and 3.2%) than in B2 (0.02%, 0.07%, 0.03%, 0.03%, 0.05%, 0.01%, 0.01%, 0.04% and 0.01%). *Gibberella* and *Meyerozyma* were not detected in B4, and *Golubevia* was not detected in B2 (Figure 5b and Appendix A).

### 3.5. Enrichment of Bacterial Colonies

Analysis of the gene pathways showed general agreement between the two bacterial groups in terms of enrichment, with the pathways with higher relative abundance being Amino Acid Biosynthesis, Cofactor, Prosthetic Group, Electron Carrier, Vitamin Biosynthesis, and Nucleoside and Nucleotide Biosynthesis (Biosynthesis) (Figure 6). In fungi, the tumor gummosis group was mainly enriched in Nucleoside and Nucleotide Biosynthesis (Biosynthesis), Electron Transfer and Respiration (all of which generate precursor metabolite and energy), and the gummosis group was mainly enriched in Cofactor, Prosthetic Group, Electron Carrier, and Vitamin Biosynthesis, Fatty Acid and Lipid Biosynthesis and Nucleoside and Nucleotide Biosynthesis (all of Biosynthesis) (Figure 7), which results suggested that the shift in fungal function may have made the gummosis worse.

## 4. Discussion

The presence of pathogens and beneficial microorganisms in plant tissues is crucial to plant health. We studied microbial composition and diversity in tumor gummosis and gummosis tissues of healthy and diseased Chinese cherry trees using Illumina MiSeq sequencing. Different microbial communities were identified in tumor gummosis and gummosis tissues. In tumor gummosis, bacterial diversity was slightly higher in B1 than in J1, while fungal diversity was slightly lower in B2 than in J2. In gummosis, bacterial diversity was lower in B3 than in J3, and fungal diversity was higher in B4 than in J4. These findings differed from those of a previous study by Li et al. [10] on *P. avium* but aligned with the tumor gummosis phase.

After analyzing diversity indices, we observed that Proteobacteria dominated the diseased tissue, which aligns with previous studies [47]. In our study, we showed Proteobacteria were much more abundant in B3 than B1. Proteobacteria was usually considered to be the main predominant colonizer in the diseased tissue, and our results support this finding. However, Firmicutes (most abundant in B1) and Actinobacteria were much more abundant in tumor gummosis (B1) than gummosis (B3). Currently, Firmicutes is also a common phylum in diseased tissue of various plants [48], and it is possible that sweet cherry gummosis was contributed. Actinobacteria, known for antibiotic production, likely play a crucial role in maintaining bacterial ecosystems [49]. Bacteroidetes showed minimal variation among samples, suggesting stability within the microbial ecosystem [50]. However, it is not clear whether the abundance of the bacteria Proteobacteria, Firmicutes, Bacteroidetes, Actinobacteria, Chloroflexi and Acidobacteria in B1 and B3 was responsible for the prevalence of the disease or whether the disease of sweet cherry causes the increase in abundance of these bacteria. The causal relationship and interaction between them merit further study.

Ascomycota was the dominant fungal phylum, and this abundance of Ascomycota was higher than reported in a previous study [51]. Mortierellomycota and Mucoromycota were detected in B2 but not in B4 and J4, and Mucoromycota was absent in J2. During the gummosis phase, fungal abundance was significantly lower in diseased tissues compared to healthy tissues. However, during the gummosis phase, fungal abundance in diseased tissues was significantly higher, indicating potential imbalances associated with gummosis disease. These findings offer insights for identifying candidates for disease biological control. Mucoromycota and Mortierellomycota are present in advanced gummosis stages, suggesting their role in exacerbating cherry runner onset.

The abundance of major microorganisms in J1 and B1 was different at the genus level. These results showed that seven bacterial genera, including *Azospirillum*, *Aquabacterium*, *Alistipes*, *Ruminococcaceae_UCG-014*, *Bacteroides*, *Clostridiales_vadinBB60_group* and *Christensenellaceae_R-7_group* were more abundant in B1 than J1. *Halomonas*, *Pelagibacterium* and *Chelativorans* were more abundant in B3 than J3. Among these, *Aquabacterium* and *Alistipes* have been widely reported to be closely related to the occurrence of disease tissue. Therefore, the bacterial genera are likely to be potential factors responsible for the gummosis of sweet cherry trees. It is worth noting that some beneficial bacteria with high abundance were present in the diseased tissue, including *Azospirillum*, *Halotolerant* and *Pelagibacterium*. It is reported that *Azospirillum* promotes the growth of plants by altering the morphology of the root system, making it a rhizosphere-promoting bacterium, and plant inoculants frequently resulted in acceleration of seed germination [52,53,54,55]. *Halotolerant* is a potential plant growth-promoting bacterium [56]. *Pelagibacterium* could degrade toxins commonly found in cereals [57]. Therefore, we hypothesized that some microorganisms in tumor gummosis may produce substances conducive to the growth of beneficial bacteria. This would allow these beneficial fungi to become the dominant microorganisms and participate in inhibiting the pathogen of gummosis disease. Further studies are needed to determine if these beneficial fungi genera are associated with the pathogen of gummosis disease in sweet cherry.

The fungal genera *Penicillium*, *Alternaria*, *Gibberella*, *Cystobasidium*, *Rhodotorula*, *Malassezia*, *Golubevia*, *Paraconiothyrium* and *Aureobasidium* were more abundant in B4 than in J4. However, the *Rhodotorula* were more abundant in B2 than in J2. *Meyerozyma* was also found in J2, while it was not found in B4, J4 and B2, and *Didymella* is highly abundant in healthy tissues. After years of research, it was found that secondary metabolites of the *Penicillium* genus have a variety of biological activities [58]. *Penicillium crustosum* is known to cause fruit rot in sweet cherries [59], while *Penicillium purpurogenum* can produce toxins that inhibit plant growth and lead to the death of *Atractylodes* [60]. *Penicillium* spp. are known as seed-borne fungal pathogens of onions that occur on onions in the field and during transit or storage [61], and *Penicillium brasilianum* could infect both the inner layers and outer surfaces of onion bulbs and cause typical symptoms [62]. *Alternaria panax* was associated with leaf spot and blight of araliaceous plants [63], and *Alternaria* spp. can cause Rice Spike Rot, Cherry Leaf Spot, and other plant diseases [64]. *Cystobasidium* spp. and *Rhodotorula* spp. are capable of breaking down inert organic macromolecules such as cellulose and lignin, which makes it possible for them to inhabit carbon sources that are challenging for bacteria to decompose [65]. Those fungal genera (particularly *Penicillium*, which was also in high abundance in B4 with 50.53% and 0.13% in J4) may be associated with gummosis disease and *Didymella* is highly abundant in healthy tissues (J4). In addition, the difference in abundance of fungi between J2 and B2 was greater. Feng et al. [66] also reported that the community composition of fungi is more sensitive to gummosis disease. Therefore, we hypothesized that the imbalance of fungal flora could be linked to the occurrence of gummosis disease in *P. avium*. In conclusion, the high abundance of bacterial genera *Halomonas*, *Pelagibacterium*, *Chelativorans*, *Pantoea*, *Aquabacterium* and *Alistipes*, and fungal genera *Penicillium*, *Cystobasidium* and *Rhodotorula* during the gummosis period is closely associated with the gummosis disease of sweet cherry. Additionally, the fungal genera *Didymella*, *Aureobasidium*, *Mycosphaerella* and *Meyerozyma* are potential antagonists of the gummosis disease. Furthermore, an imbalance of fungal flora could potentially cause sweet cherry disease. Although further research is necessary to confirm these findings, this study represents the first step in identifying candidates for the biological control of gummosis in cherry trees.

In our study, we focused on analyzing the tumor gummosis in diseased cherry trees and the microbial composition and diversity within the gummosis tissues. The results showed that a variety of microbial communities were present in the diseased tissues, which may have played a significant role in the development of the disease. However, beyond the direct impact of the microbes themselves, we must also consider the indirect effects of other environmental factors on the disease. In recent years, an increasing number of studies have shown that the interactions between plants, insects and microbes play a critical role in plant diseases [67]. Soil conditions, insect activity and the synergistic effect with pathogenic microbes may have played a crucial role in the occurrence and development of tumor gummosis in cherry trees [68]. For instance, during the cultivation of cherry trees in Bailuyuan, high soil particle viscosity and poor aeration, combined with the activity of the underground insect *Maladera verticalis*, facilitated the invasion of pathogenic microbes [69]. The physical damage caused by insects not only weakened the tree’s vigor but also opened up pathways for microbial invasion [70]. The rampant damage caused by the *Aromia bungii* further weakened the health of cherry trees, exacerbating the severity of the disease [71]. Therefore, our study not only focuses on the direct impact of microbes but also suggests that the cause of the disease should be understood from a systemic perspective. By comprehensively considering the interactions between soil, insects and microbes, a more complete understanding of the complexity of cherry tree tumor gummosis can be achieved, providing theoretical support for the development of more effective control strategies [72].

## 5. Conclusions

A comprehensive understanding of the possible involvement of microorganisms in the development and progression of gummosis can be gained from a comparative study of the microbial composition and diversity in tumor gummosis and gummosis tissue of healthy and diseased cherry trees. Differences in the bacterial and fungal composition of healthy and diseased tissue were discovered, suggesting that imbalances in the microbial community may play a role in the development of the disease. The major bacterial and fungal phyla in diseased tissue were Proteobacteria and Ascomycota, respectively. In addition, some beneficial bacteria and fungi were found to be more prevalent in the damaged tissue, suggesting that they may play a role in controlling the pathogen causing gummosis.

## Figures and Tables

**Figure 1 microorganisms-12-01837-f001:**
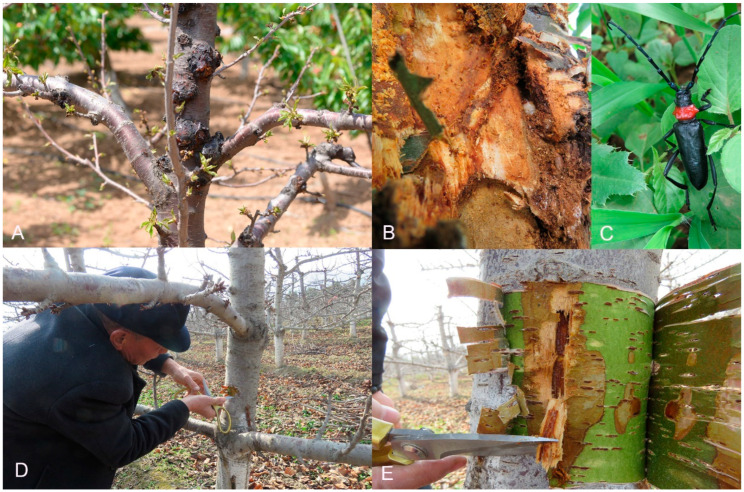
Symptoms of sweet cherry stem disease. (**A**) Gummosis includes protruding xylem and black nodular spots in Bailuyuan; (**B**) Larva of the longhorn beetle *Aromia bungii* in the tumor of stem; (**C**) the longhorn beetle *Aromia bungii*, adult (**D**,**E**) Gummosis of the trunk and branches in Mayuhe.

**Figure 2 microorganisms-12-01837-f002:**
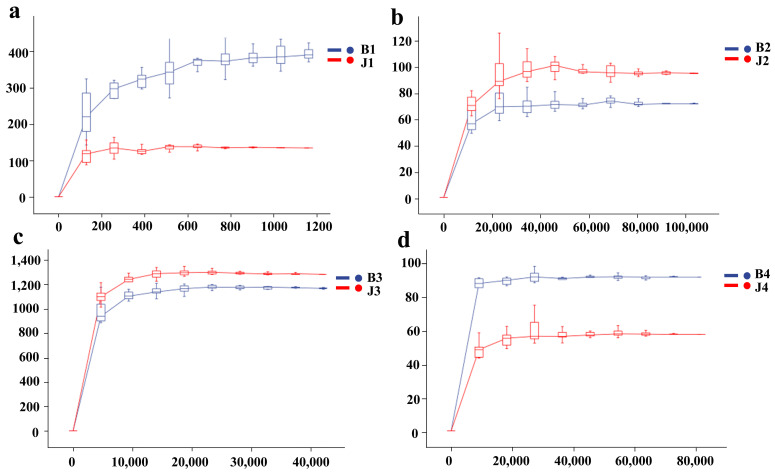
Sparse curves of the number of observed species of bacteria and fungi in the disease and the healthy samples. (**a**) B1/J1; (**b**) B2/J2; (**c**) B3/J3; (**d**) B4/J4. Note: In bacteria groups, tumor gummosis (B1) and healthy tissue (J1) at Bailuyuan; gummi (B3) and healthy tissue (J3) at Mayuhe. In fungi groups, tumor gum (B2) and healthy tissue (J2) at Bailuyuan; gummi (B4) and healthy tissue (J4) at Mayuhe.

**Figure 3 microorganisms-12-01837-f003:**
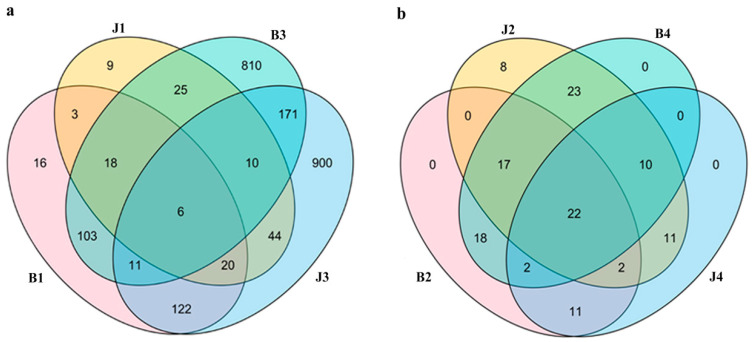
(**a**) Bacterial and (**b**) fungal Venn diagrams at the operation taxonomic units (ASVs). Venn diagrams showing the number of shared and unique ASVs (≥100 similarity) among tissues. (Online coloration). Note: In bacteria groups, tumor gummosis (B1) and healthy tissue (J1) at Bailuyuan; gummi (B3) and healthy tissue (J3) at Mayuhe. In fungi groups, tumor gummosis (B2) and healthy tissue (J2) at Bailuyuan; gummi (B4) and healthy tissue (J4) at Mayuhe.

**Figure 4 microorganisms-12-01837-f004:**
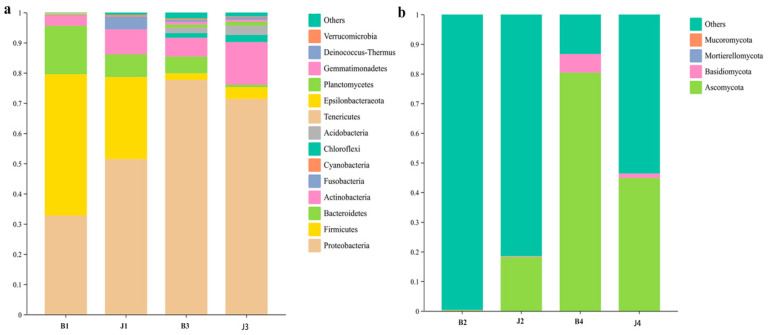
Relative abundance of the dominant groups of bacteria (**a**) and fungi (**b**) at the phylum level in tissue samples from healthy and tumor gummosis tissues, healthy and tumor gummosis tissue. The longer the column, the higher the relative abundance of the taxon in the corresponding sample. (Online coloration). Note: In bacteria groups, tumor gummosis (B1) and healthy tissue (J1) at Bailuyuan; gummi (B3) and healthy tissue (J3) at Mayuhe. In fungi groups, tumor gummosis (B2) and healthy tissue (J2) at Bailuyuan; gummi (B4) and healthy tissue (J4) at Mayuhe.

**Figure 5 microorganisms-12-01837-f005:**
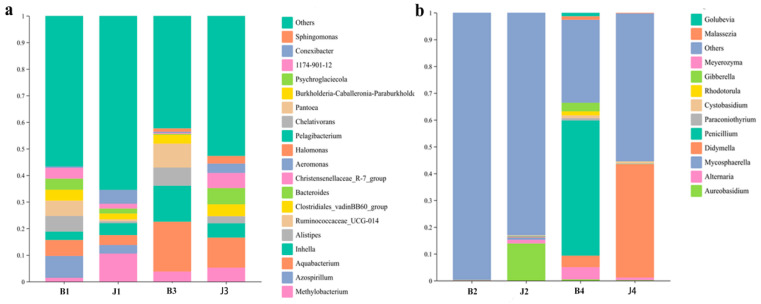
(**a**) Bacterial and (**b**) fungal relative abundance of the dominant group at the genus level. The longer the column, the higher the relative abundance of the taxon in the corresponding sample (Online coloration). Note: In bacteria groups, tumor gummosis (B1) at Bailuyuan; gummi tissue (B3) at Mayuhe. In fungi groups, tumor gummosis (B2) and at Bailuyuan; gummi tissue (B4) at Mayuhe.

**Figure 6 microorganisms-12-01837-f006:**
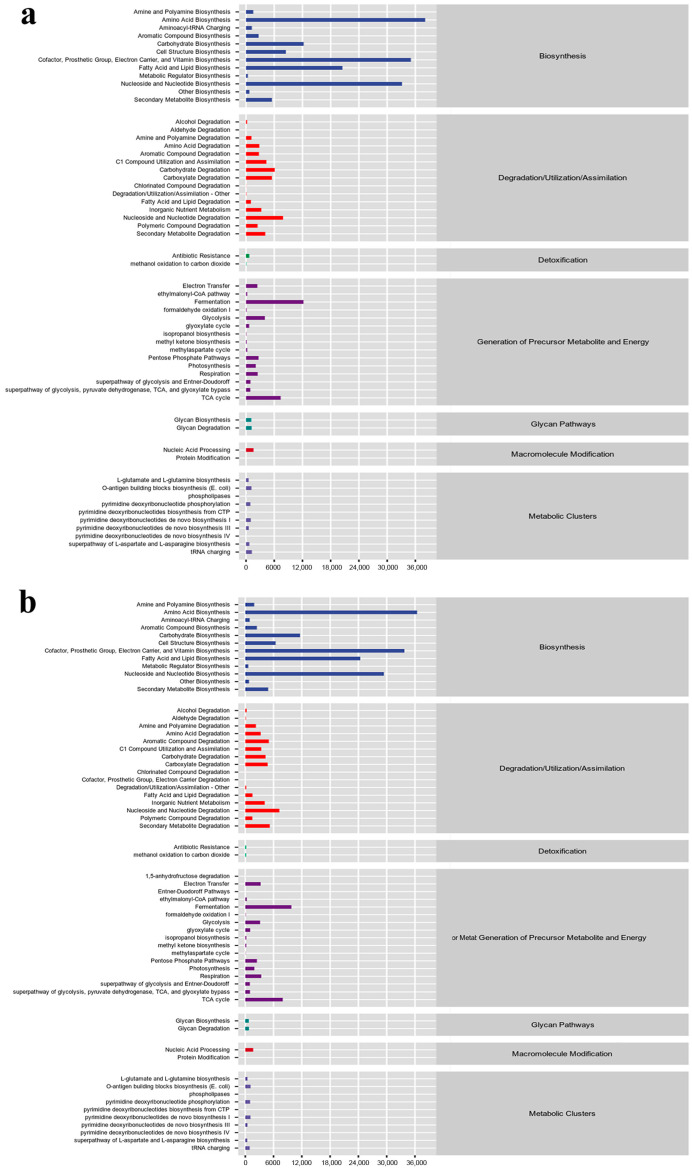
The predictive function of tissue bacteria during (**a**) tumor gummosis and (**b**) gummosis tissue by relative abundance.

**Figure 7 microorganisms-12-01837-f007:**
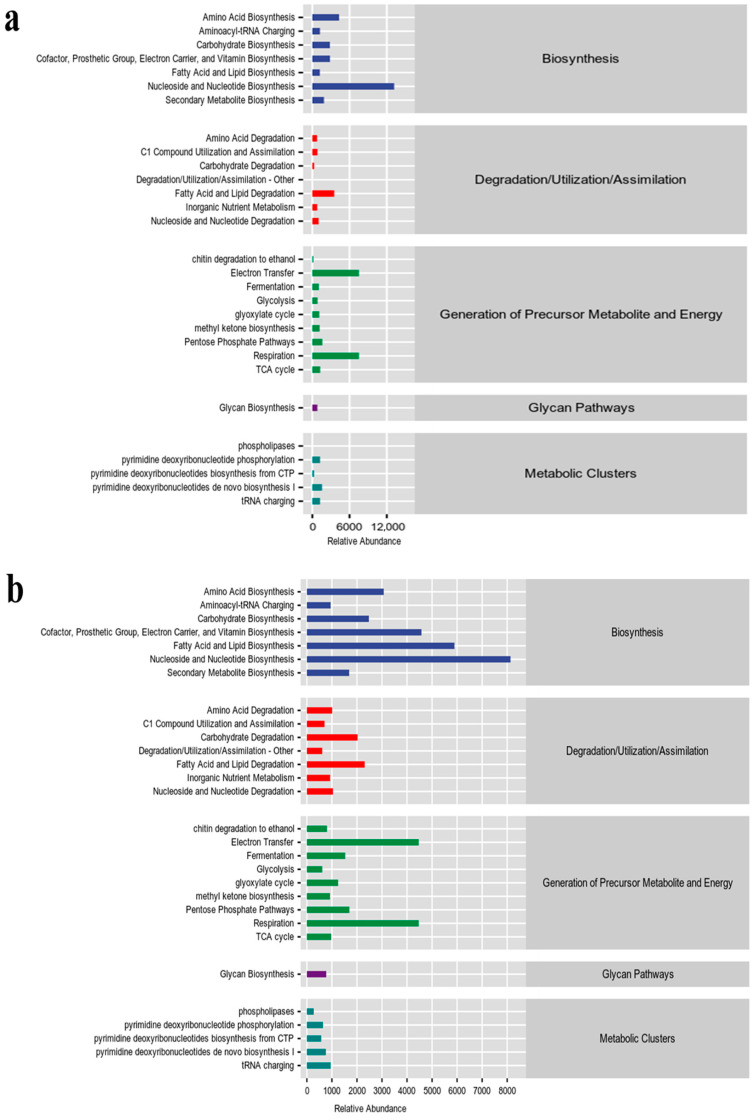
The predictive function of tissue fungi during (**a**) tumor gummosis and (**b**) gummosis tissue by relative abundance.

**Table 1 microorganisms-12-01837-t001:** Alpha diversity of microbial communities in tissue samples from healthy and gummosis-diseased sweet cherry trees.

Sample	Simpson	Chao1	Shannon	Goods_Coverage	Observed_Species	Pielou_e	Faith_pd
J1	0.979	135.2	6.27	0.996735	135	0.886	14.53
B1	0.982	394.8	7.09	0.900172	295	0.864	21.99
J2	0.881	95.3	3.48	0.999970	95	0.530	-
B2	0.859	72.1	3.15	0.999983	72	0.510	-
J3	0.971	1284.9	6.98	0.999516	1284	0.676	91.05
B3	0.931	1171.8	6.17	0.998383	1155	0.607	83.75
J4	0.710	58.0	2.22	0.999995	58	0.379	-
B4	0.723	92.1	3.10	0.999996	92	0.476	-

Note: Shannon and Simpson: representational diversity; Chao1: representational richness; Good’s coverage: representation coverage; Pielou’s e (Pielou’s evenness): Exponential characterization of homogeneity; Faith’s PD: Characterizing evolutionary-based diversity. And in bacteria groups, tumor gummosis (B1) and healthy tissue (J1) at Bailuyuan; gummi (B3) and healthy tissue (J3) at Mayuhe. In fungi groups, tumor gummosis (B2) and healthy tissue (J2) at Bailuyuan; gummi (B4) and healthy tissue (J4) at Mayuhe.

## Data Availability

The original contributions presented in the study are included in the article/Appendix A, further inquiries can be directed to the corresponding author.

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
