# Peer review of "Comparative Analysis of Microbial Communities in Diseased and Healthy Sweet Cherry Trees (Prunus avium L.)"

_microorganisms, 2024, doi:10.3390/microorganisms12091837_

Round 1

Reviewer 1 Report

Comments and Suggestions for Authors

 -          The title  reflects the paper content

-          This work represents a significant contribution to development of biological control of the disease (gummosis) on the base of identification pathogenic microorganisms responsible for gummosis disease, which rapidly spread in the most popular and economically valuable fruits European sweet cherry Prunus avium (L.) belongs to the Rosaceae family on the base of research at the garden of Bailuyuan which heavily suffered by gummosis disease and horn beetle damage with the orchard of Zhouzhi which only with gummosis disease, both from Shaanxi, China.

-          The presented results represents a significant contribution by determination on the level of phylum and genera of different bacterial communities as well as fungal communities in tumor gummosis and gummosis tissues and their imbalance in gummosis disease tissues. On the base of  analysis were identified some bacterial genera Pelagibacterium, Halomonas, Azospirillum, Aquabacterium and Alistipes) and fungal genera (Penicillium, Alternaria and Rhodotorula) in the diseased tissues of gummosis. Among these, the increased relative abundance of the bacteria genes Halomonas, Pelagibacterium, Chelativorans, Pantoea, Aquabacterium, Alternaria and fungi genes Penicillium, Cystobasidium, Rhodotorula may be associated with gummosis of Prunus avium. The bacterial genera Methylobacterium, Psychroglaciecola, Aeromonas, Conexibacter and fungal genera Didymella, Aureobasidium, Mycosphaerella, Meyerozyma are probably antagonists of the pathogen of gummosis.

-          The main question of research is focused on analyzing the tumor gummosis in diseased cherry trees and the microbial composition and diversity within the gummosis tissues.

In the  study of the tumor gummosis in diseased cherry trees identified  variety of microbial communities were present in the diseased tissues and estimated their role in the development of the disease, as well as effects of other environmental factors on the disease development.

 -          Key words are appropriate.

 -          The aim of study  is clear, but  should be write as particular  paragraph on the end of chapter of Introduction.

Accordingly, should be delete  the text from lines 84 to 87 does not represent aim of study.

 -          Results are clearly presented and discussed.

 -          Tables, figures, pictures are clear.

-          The Conclusion is clear, but there is room for improvement based on the concrete results achieved in these researches!.

 -          Manuscript is acceptable after minor corrections.

Suggestion:

The text from lines 84 to 87 does not represent aim of study. Should be deleted text  “These results should offer a better understanding ofthe microorganisms that cause gummosis disease in sweet cherry trees. Additionally, this lays the foundation for further research on the biology of gummosis and provides potential biocontrol strains for achieving high quality and yield”.

In line 285 was written twice word “…..were were….”. Delete one word “were” !

 The text from lines 422 to 426 does not represent a conclusion based on research results. Should be deleted. “However, further research is needed to establish causal relationships and understand the temporal dynamics of the microbial community in gum disease. Longitudinal studies to monitor changes in microbial composition over time would provide a more comprehensive understanding of disease progression and the specific role they play different microorganisms in different stages.”

-Abbreviations should be listed before the introduction or according to the editor's suggestions.

Author Response

Comments 1: The text from lines 84 to 87 does not represent aim of study. Should be deleted text “These results should offer a better understanding of the microorganisms that cause gummosis disease in sweet cherry trees. Additionally, this lays the foundation for further research on the biology of gummosis and provides potential biocontrol strains for achieving high quality and yield.

Response 1: Thank you for your insightful suggestion. We agree that the text in lines 84 to 87 could be misleading in terms of representing the study's aim. As per your recommendation, we have deleted this section. The revised paragraph now more clearly aligns with the overall objective of the study.

“In recent years, our research group found that the sweet cherry in Bailuyuan (loess plateau) affected by "black knot disease" were also severely damaged by the stem borer, the red-necked longhorn beetle Aromia bungii [29] (Coleoptera: Cerambycidae). In the orchard on the sandy soil near the foot of Mount Qin, the sweet cherry tree with gummosis syndrome, but without the longhorn beetle Aromia bungii damage did not have tumors. Obviously, comparing the microbial communities of the diseased tissues from these two areas is possible to make the main cause clear. The objective of this study used Illumina sequencing technology to investigate the differences between bacterial and fungal communities in healthy and diseased tissues of the sweet cherry. These results should offer a better understanding ofthe microorganisms that cause gummosis disease in sweet cherry trees. Additionally, this lays the foundation for further research on the biology of gummosis and provides potential biocontrol strains for achieving high quality and yield. ”

Comments 2: In line 285 was written twice word “…..were were….”. Delete one word “were” !

Response 2Thank you for pointing out the repetition. We have corrected the error by removing the duplicate ‘were’ in line 285. The revised text now reads correctly.

“Mortierellomycota and Mucoromycota were were not detected in B4. ”

Comments 3: The text from lines 422 to 426 does not represent a conclusion based on research results. Should be deleted. “However, further research is needed to establish causal relationships and understand the temporal dynamics of the microbial community in gum disease. Longitudinal studies to monitor changes in microbial composition over time would provide a more comprehensive understanding of disease progression and the specific role they play different microorganisms in different stages.”

Response 3Thank you! We have deleted the section.

“A comprehensive understanding of the possible involvement of microorganisms in the development and progression of gummosis can be gained by a comparative study of the microbial composition and diversity in tumor gummosis and gummosis tissue of healthy and diseased cherry trees. Differences in the bacterial and fungal composition of healthy and diseased tissue were discovered, suggesting that imbalances in the microbial community may play a role in the development of the disease. The major bacterial and fungal phylum in diseased tissue were Proteobacteria and Ascomycota, respectively. In addition, some beneficial bacteria and fungi were found to be more prevalent in the damaged tissue, suggesting that they may play a role in controlling the pathogen causing gummosis. However, further research is needed to establish causal relationships and understand the temporal dynamics of the microbial community in gummosis disease. Longitudinal studies to track changes in microbial composition over time would provide a more comprehensive understanding of the disease progression and the specific roles played by different microorganisms at various stages.

Comments 4: Abbreviations should be listed before the introduction or according to the editor's suggestions.

Response 4Thank you for the suggestion regarding the listing of abbreviations. We have reviewed our manuscript and found that the abbreviations used are ‘L.’ for Linnaeus and ‘16S rRNA’ for 16S ribosomal RNA. Both are standard and widely recognized in botanical and microbiological nomenclature. Since these abbreviations are commonly understood in their respective contexts, we have not included a separate list of abbreviations. We appreciate your understanding and the attention to detail.

Reviewer 2 Report

Comments and Suggestions for Authors

The manuscript with the title “Comparative Analysis of Microbial Communities in Diseased 2 and Healthy Sweet Cherry Trees (Prunus avium L.)” screened for fungal and bacteria from healthy/diseased cherry orchards (affected by gummosis and horn beetle and respectively affected only by gummosis). The results are promising because they hint to potential biological agents of disease and pest control for cherry.

The introduction is providing a very good background as well as motivation for the research.

Material and Method

What are the general climatic conditions in the areas where these 2 orchards were located?

Is it anything known about the cherry trees in the two locations, such as if they have some susceptibility or genetic resistance to these diseases or pest attack? Such as specific cultivars.

2.1. could authors detail what the collected tissue consisted of? Bark, wood etc. from diseased area on the trunk... Please if possible to be more specific of the type of tissue contained in the collected samples, and how deep into the bark were collected.

Results

Figures 6 and 7 could be enlarged.

Were the samples from experimental variants containing the same type of tissues? Because even in healthy plants the bacteria and fungal colonizers probably have affinity for certain tissue types and layers. Could authors say anything about this aspect?

It would have been interesting to have some chemical analysis of the tissue. But the results are complex nonetheless.

Best regards.

Comments on the Quality of English Language

English style and grammar need moderate revision. Also, please revise syntax where necessary.

Author Response

Comments 1: What are the general climatic conditions in the areas where these 2 orchards were located?

Response 1: Thanks. We add the detail GPS information of the two orchards. The distance between them only 58.7 km, both at the Northern Piedmont of Mt. Qinling. The orchard of Bailuyuan is 200 m higher than that of Mayuhe. Probably Bailuyuan is 1.2︒C cooler than Mayuhe.

Comments 2: Is it anything known about the cherry trees in the two locations, such as if they have some susceptibility or genetic resistance to these diseases or pest attack? Such as specific cultivars.

Response 2: It is an interesting question. We have insulted the farmer. In the two orchards, they planted the cherry trees using the same rootstock named Gisela which showing the best genetic resistance. According to our investigated, the “black knot disease” is popular in Bailuyuan and no distinct difference among the varieties. Soil permeability might be the main reason which make the tree growing difference. Pathogenic microorganisms and horn beetle attack the weak tree firstly. We focus on compare the pathogenic microorganisms in this paper.  

Comments 3: 2.1. could authors detail what the collected tissue consisted of? Bark, wood etc. from diseased area on the trunk... Please if possible to be more specific of the type of tissue contained in the collected samples, and how deep into the bark were collected.

Response 3: Thank you for your kind comments. For samples collection, we followed Ding et al. (2021). The diseased and healthy inner bark epidermal tissues were collected using a five-point sampling method. Each sampling point was sampled three times and mixed into one sample, and then the sample was ground and passed through a 2-mm sieve. The samples were collected in a sterile container and stored at 4°C and DNA extraction was performed within 48 hours.

Comments 4: Figures 6 and 7 could be enlarged.

Response 4: Thanks. We have enlarged the pictures.

Comments 5: Were the samples from experimental variants containing the same type of tissues? Because even in healthy plants the bacteria and fungal colonizers probably have affinity for certain tissue types and layers. Could authors say anything about this aspect? It would have been interesting to have some chemical analysis of the tissue. But the results are complex nonetheless.

Response 5: Sorry, we have not done the comparation among different tissues. It is really a nice question and we would do it in future.

Comments 6: English style and grammar need moderate revision. Also, please revise syntax where necessary.

Response 6: Thank you for your suggestion. We have carefully reviewed the grammatical errors and made adjustments to certain parts of the manuscript. We have made every effort to ensure that the English style and grammar of the text meet the standard. If there are any additional areas that need further revision, please let us know, and we will continue to make improvements.